# Relations among Poly-Bullying Victimization, Subjective Well-Being and Resilience in a Sample of Late Adolescents

**DOI:** 10.3390/ijerph17020590

**Published:** 2020-01-16

**Authors:** Beatriz Víllora, Elisa Larrañaga, Santiago Yubero, Antonio Alfaro, Raúl Navarro

**Affiliations:** 1Department of Psychology, Faculty of Education and Humanities, University of Castilla-La Mancha, Avda de los Alfares, 42, 16071 Cuenca, Spain; elisa.larranaga@uclm.es (E.L.); Santiago.Yubero@uclm.es (S.Y.); 2Department of Pedagogy, Faculty of Education and Humanities, University of Castilla-La Mancha, Avda de los Alfares, 42, 16071 Cuenca, Spain; antonio.alfaro@uclm.es

**Keywords:** bullying, subjective well-being, resilience, poly-victimization, late adolescents

## Abstract

The present study examined the relations among poly-bullying victimization (experiencing multiple forms of peer bullying), resilience and subjective well-being. This study specifically examined late adolescents’ resilience as a moderator of the relation between poly-bullying victimization and subjective well-being. In a region of central Spain, 1430 undergraduate students (64% females, 36% males), aged between 18 and 22 years, completed three self-reported measures, including bullying victimization experiences, self-reported subjective well-being and resilience. A substantial proportion of the participants (16.9%) reported being victims of poly-bullying. The results showed that the poly-bullying victimization group reported the poorest subjective well-being and the lowest resilience levels. The regression analyses revealed that resilience was significantly and positively associated with subjective well-being, and resilience moderated the association between poly-bullying victimization and subjective well-being. However, the relation was very weak and accounted for only an additional 1% of variance in the participants’ subjective well-being. Future research should assess resilience trajectories of youth exposed to multiple forms of bullying victimization in order to better understand the potential protective effect of resilience over negative mental health outcomes.

## 1. Introduction

There is a general consensus among researchers about the importance of the social and psychological well-being of relationships between peers in adolescence and emerging adulthood [1]. However, when these relationships are characterized by bullying or aggression, they have negative consequences for physical and mental health. During this development period, youths may be more susceptible to bullying victimization effects due to changes in experienced social relationships, intense emotions and biological changes [2]. Specifically, the body changes related with the onset of adolescence and adulthood and changes in friendships availability, individuals’ status within groups in secondary schools and universities, and peer support accessibility are factors associated with victimization experiences.

Olweus [3] explained that bullying occurs when “a person is exposed, repeatedly and over time, to negative actions on the part of one or more other persons”. Bullying is a peer aggressive behavior characterized by persistence, intentionality and power imbalance between victims and perpetrators [4]. Several bullying victimization types have been identified [5,6,7]. The most frequent categories are physical, verbal, indirect or social/relational, and cyberbullying. Physical aggression includes hitting, kicking, punching, taking or damaging belongings; verbal aggression comprises teasing, taunting or threatening; relational/social aggression or social exclusion includes rumors or exclusion with the intention to damage self-esteem and/or social status [6]. Cyberbullying consists of using Information and Communication Technologies (ICTs; email, instant messaging or “chat”, websites or blogs, online video games and mobile phones) to exercise psychological abuse [8].

Wang, Iannotti, Luk and Nansel [7] analyzed the prevalence rates of different bullying victimization types in a nationally representative sample of adolescents from the USA. The prevalence rates of involvement in the five victimization types were 13.2% for physical, 36.9% for verbal, 25.8% for social exclusion, 32.1% for rumor spreading and 10.1% for cyberbullying. In Korea, Kim and Yun [9] obtained a prevalence rate of 3.5% for physical bullying victimization, 12.8% for verbal bullying victimization, 18.1% for social/relational bullying victimization, 5.6% for cyberbullying and 5.5% four poly-bullying victimization. In Spain, the percentages of the prevalence rates of physical, verbal, and social/relational bullying are 4.1%, 14.6% and 13.1%, respectively [10]. In a review of 14 studies, Lund and Ross [11] found an average prevalence of approximately 20–25% of victimization in university students. Regarding victimization types, Pontzer [12] reported prevalence rates of 15.5% for verbal victimization, 2.1% for physical victimization and 8% of social/relational victimization. Wensley and Campbell [13] reported a prevalence rate of 11.6% for cyberbullying and 20.8% for traditional bullying.

Bullying victimization, that is both “face to face”, and through the ICT, includes a wide range of serious short-term and long-term mental health problems [14,15,16,17,18,19,20,21]. A quasi-experimental meta-analytical study showed how bullying victimization generates behavioral and emotional problems as symptoms of their psychological distress in response to the bullying experience, including internalizing symptoms such as anxiety, depression and suicidality [22]. Bullying victimization is also associated with unmet need for mental health treatment and counseling [23]. Longitudinal studies have revealed that bullying and cyberbullying victimization experiences in adolescence have a significant impact on mental health outcomes later in adulthood [16,17].

Research on victimization experiences has grown over the years. However, much of this research has focused on specific victimization types and their impact on victims [24]. More recently, research has paid more attention to the impact of exposure to multiple victimization types, or “poly-victimization” [25]. Dierkhising, Ford, Branson, Grasso and Lee [26] defined poly-victimization as “exposure to multiple and varied traumatic stressors involving interpersonal violence” (p. 41). It has been found that different forms of victimization are interrelated in such a way that the adolescents who experience one type are also likely to be exposed to other forms of victimization [27].

Exposure to multiple types of violence and victimization has immediate and lifelong effects on physical and mental health [28,29,30]. Previous research showed that poly-victimization represents a substantial source of higher mental health risk than exposure to any individual victimization type, even when it repeatedly occurs [27,28,29,30,31].

Hickman et al. [32] measured violence exposure in three different ways: (1) total frequency of all lifetime exposure; (2) total frequency of lifetime exposure by a broad category (measuring violence according to each category, i.e., assault, maltreatment, sexual abuse, witnessing violence); (3) poly-victimization (exposure to multiple violence categories). The results indicated that only poly-victimization emerges as a consistent predictor of negative symptoms, such as posttraumatic stress disorder and behavioral problems.

Experiencing multiple forms of peer bullying has also been simultaneously found to negatively affect youths’ psychological and social well-being. For example, Raskauskas [33] found that students who experienced more than one form of peer victimization also report more severe depressive symptoms, lower self-esteem, and are more likely to use self-blame causal attributions to explain their own victimization. The students who have experienced two victimization types or more obtain worse mental health scores than those who have experienced only one form or were not involved. Along the same line, Wigderson and Lynch [34] found that multiple forms of peer victimization are negatively associated with adolescent well-being, and victimization is positively associated with self-reported anxiety and depression, and inversely associated with self-esteem.

The primary aim of the present study was to examine the relations among poly-bullying victimization (experiencing multiple forms of peer bullying), resilience and subjective well-being. Although bullying occurs in different settings (e.g., educative centers, prisons and workplaces) and may involve different individuals (e.g., siblings, students, workplace mates), our analyses will be center on bullying occurring between students in the university context. As it was described above, bullying behaviors between students can take various forms, including physical behaviors (hitting, kicking), verbal attacks (name calling, threats), social/relational aggression (excluding someone from a group, rumor spreading), and cyberbullying (posting someone’s private photos online). Our study examines these four types of bullying between students.

### 1.1. Subjective Well-Being and Bullying

The World Health Organization (WHO) defines mental health as a positive state, “a state of well-being in which the individual realizes his or her own abilities, can cope with the normal stresses of life, can work productively and fruitfully, and is able to make a contribution to his or her community” [35] (p. 12).

Defining mental health positively implies conceiving mental health as not only the absence of mental illness but also as the presence of different psychosocial resources that contribute to the realization of one’s full individual potential [36].That is to say, positive mental health is related to, but in a different form, mental illness [37]. Therefore, an assessment of positive mental health significantly supplements the assessment of mental illness [38]. Keyes [37] operationalized positive mental health through measures of subjective well-being by including three dimensions. Emotional well-being [39] reflects the presence of positive affect, the absence of negative affect and satisfaction with one’s life. Psychological well-being [40] includes self-acceptance, personal growth, purpose in life, positive relationships with others, autonomy and environmental mastery. Social well-being [41] is integrated by social integration, social contribution, social coherence and social acceptance, and focuses on the evaluations of individuals in their public and social lives.

Previous research has suggested that bullying and cyberbullying negatively impact victims’ subjective well-being [42]. With a sample of preadolescents, Navarro, Ruiz-Oliva, Larrañaga and Yubero [43] found that cyberbullying and social bullying victims report less subjective well-being than uninvolved children. Along the same line, Savahl, et al. [44] found a negative relation between subjective well-being and physical and social/relational bullying in a sample of children from 15 countries. These same results have also been found for adolescents, late adolescents and emerging adults. For example, González-Cabrera, Machimbarrena, Ortega-Barón, and Álvarez-Bardón [45] reported that youths victimized through mixed venues (traditional victims and cyberbullying victimization) have a lower health-related quality of life compared to those victimized only through either traditional bullying or cyberbullying. Baier and Kunkel [46] found that social/relational bullying negatively affects adolescents’ psychological well-being. Chen and Huang [47] demonstrated that traditional victimization among pre-university students and university students is negatively associated with well-being.

The same relations have also been found in poly-victimization research. Víllora, Yubero and Navarro [48] observed how university students experiencing poly-victimization (traditional bullying and cyber dating abuse) report lower levels of emotional, social and psychological well-being than non-involved students or those suffering only cyber dating abuse. Mitchell, Moschella, Hamby and Banyard [49] reported a linear decline in subjective well-being, mental health and the number of healthy days as victimization becomes more persistent across childhood and more diverse in forms of victimization. Adolescents and adults who have been poly-victimized in more than one child development stage display the lowest well-being levels.

### 1.2. Resilience and Subjective Well-Being

Although research has shown that victimization experiences have short-term and long-term negative effects on mental health, not all victims experience the same outcomes. Differences in outcomes may result from different resilience levels. Indeed, the term resilience is used to “refer to the finding that some individuals have a relatively good psychological outcome despite suffering risk experiences that would be expected to bring about serious sequelae” [50] (p. 1). Resilience is linked to exposure to severe adversity and to achieving positive adaptation despite this significant threat [51].

Resilience is understood as a personal trait that is inherent to an individual, but also as a process or phenomenon influenced by the cultural and social context [52]. The factors contributing to resilience have been conceptualized as both cognitive-individual (self-regulation, planning, executive functioning, problem-solving skills and self-efficacy) and social-interpersonal (social skills, social support and quality of family relationships) [53]. These factors serve as endemic resources to help youths to believe, through experience, that they can face and overcome various stressors in their lives [54]. In a meta-analysis, resilience was found to correlate negatively with negative affect, anxiety and depression, but positively with positive indicators of mental health, positive affect and life satisfaction. The authors of this analysis suggested that resilience plays a major role in helping individuals to achieve a state of positive mental health and to reduce negative indicators [55]. More recently, Simón-Sáiz, et al. [56] reported that the most resilient adolescents obtain better results in all the quality of life dimensions, and that the effect of resilience is stronger in mental health-related dimensions.

Different studies have found that resilience is negatively related to bullying and cyberbullying, and it moderates the relation between bullying victimization and youths’ negative mental health outcomes. For example, Hinduja and Patchin [54] found that adolescents with higher resilience levels are less likely to report that they have been bullied at school or online. Zhou, Liu, Niu, Sun and Fan [57] found resilience to be an important factor that mediates the relation between bullying victimization and childhood depression. Navarro, Yubero and Larrañaga [58] indicated that resilience is negatively associated with cyberbullying victimization and fatalism. Resilience lowers the likelihood of developing fatalistic beliefs and engagement in cyberbullying among Spanish adolescents. Resilience also moderates the association between cyberbullying victimization and the development of fatalistic beliefs. Similarly, Huang and Mossige [59] indicated that resilience has a strong and significant negative association with poor mental health, and also substantially moderates the negative relation between poly-victimization and young people’s mental health. However, research analyzing the association between resilience, subjective well-being and victimization by multiples forms of bullying is lacking. Our intention was to analyze whether there is a negative relationship between poly-bullying victimization and subjective well-being among late adolescents and whether resilience moderates this potential negative relation.

### 1.3. Present Study

The present study was designed to analyze the associations among poly-bullying victimization and reports of well-being, and to explore the moderation effect of resilience on the relation between these two variables. Consequently, the purposes of this study were to: (a) explore the prevalence of poly-bullying victims among late adolescents; (b) examine the association between well-being and resilience among late adolescents exposed to multiple forms of bullying; (c) analyze the moderating effects of resilience on poly-bullying victimization and well-being. Therefore, the following hypotheses are proposed:

**Hypothesis** **1.**
*Subjective well-being will be poorer among poly-bullying victims than among non-victims and other groups of victims.*


**Hypothesis** **2.**
*Poly-bullying victimization will be related to poorer subjective well-being, whereas resilience will be related with better subjective well-being.*


**Hypothesis** **3.**
*Resilience will moderate the relation between poly-bullying victimization and subjective well-being.*


## 2. Method

### 2.1. Design and Participants

Data used in this study are from a non-probabilistic sample formed by 1430 undergraduate students from a Spanish university located in central Spain with approximately 23.000 students. Specifically, 64% of the participants were females and 36% were males. Their ages ranged from 18 to 22 years (M = 19.66; SD = 1.25). The data in the present study were collected in 2018 and involved convenience sampling from 17 faculties of the University of Castilla-La Mancha with students attending all knowledge areas in years 1 to 3. The university student population in the 2017/2018 academic year was 22.483 undergraduate students. Sample size was calculated considering a *z* value of 1.96 (95% confidence level) and ± 3.4% error margin with an expected proportion (P) of 0.5. The analyses determined that 1019 students were required for the study. We intentionally oversampled.

### 2.2. Measurement Variables and Instruments

The participants provided information about demographic variables, such as gender, age and grade. The following instruments were used to analyze the study variables.

Self-reported bullying victimization. Bullying victimization at university was assessed using “The Bullyharm” developed by Hall [60], which comprises 14 items to self-report different bullying behaviors between students (physical, verbal and social bullying) and on the Internet (cyberbullying) occurring in the last 3 months (item example for physical bullying: “During the past three months, other students pushed or pulled on me”; item example for verbal bullying: During the past three months, other students said something mean to me”; item example for social bullying: “During the past three months, other students tried to turn people against me“; item example for cyberbullying: “During the past three months, other students sent me a mean email, instant message, or text message”). Items score on a 4-point scale: 0 (never); 1 (once or twice in the past 3 months); 2 (about once a week) or 3 (twice a week or more). Internal consistency was 0.77 for the whole victimization scale.

Subjective well-being. Well-being was measured using “The Mental Health Continuum-Short Form (MHC-SF) developed by Keyes, Wissing, Potgieter, Temane, Kruger and van Rooy [61], and validated and adapted to Spanish by Echeverría, Torres, Pedrals, Padilla, Rigotti and Bitran [36]. The MHC-SF was designed to assess individuals’ perceptions of their affective states, and their psychological and social functioning. The scale comprises 14 items rated on a 6-point Likert scale based on the feelings that they had had in the last month and ranges from never to every day. Higher values indicate better subjective well-being. Internal consistency was 0.90 in the current sample.

Resilience. The Scale of Protective Factors-24 (SPF-24) [53] was used to assess resilience. The scale comprises 24 items measuring social and personal sources of resilience, such as social support, social skills, planning and prioritizing behavior. Participants’ responses indicate their level of agreement with each item on a 7-point Likert scale, ranging from “1 = completely disagree” to “7 = completely agree”. We used the scale to obtain one score for resilience by summing all the items, where a higher value indicated higher resilience levels. Cronbach’s α internal consistency reliability for the SPF was 0.92 in the current sample.

### 2.3. Procedure

Data were collected by self-reported group class-administered pencil-and-paper anonymous questionnaires. One researcher administered questionnaires to the participants and answered any questions asked by the participants. The procedure took approximately 20 min in each group class. The Clinical Research Ethics Committee of the Virgen de la Luz Hospital in Cuenca approved the study protocol (PI0519). All the subjects signed informed consent forms prior to participating in the study.

### 2.4. Analysis Plan

Based on the identification and study of poly-bullying victimization prevalence, we first analyzed the prevalence of the participants reporting bullying victimization across the four examined forms of bullying (physical, verbal, social, cyberbullying) and the groups of victims by considering the number of forms endured. Second, we compared reports of well-being and resilience across the different victimization groups by applying an analysis of variance (ANOVA). Third, a linear regression analysis was used to estimate the relations of poly-bullying victimization and resilience on subjective well-being (Step 1), and to then detect whether resilience moderated the association of poly-bullying victimization on late adolescents’ subjective well-being by introducing the interaction between resilience and poly-bullying. Statistical analyses were done using the software package SPSS 22.0.

## 3. Results

### 3.1. Prevalence of Poly-Bullying Victimization among Late Adolescents

Students were considered victims of each form of bullying if they reported having suffered one of the behaviors indicated in the questionnaire at least once a week in the last 3 months. The remaining students were considered to not be involved in bullying. Students were also classified according to the number of experienced forms of bullying (see Table 1), and 16.9% of the participants reported suffering three or four forms of bullying. Following the criteria used by previous research on poly-victimization [55], these two groups were taken as the poly-bullying victimization group.

### 3.2. Well-Being and Resilience among Late Adolescents Exposed to Several Forms of Bullying

As shown in Table 2, the female participants more often reported being bullied than males, although the difference was not statistically significant (*X*^2^ = 9.175 (3), *p* = 0.057). A one-way ANOVA was calculated for subjective well-being and resilience by considering the different groups of victims. The analysis was significant for subjective well-being [*F* (3, 1.430) = 12.243, *p* < 0.001] and resilience [*F* (3, 1.430) = 6.875, *p* < 0.001]. A follow-up analysis using Bonferroni correction showed that non-victims reported better subjective well-being and higher resilience levels than victims. The poly-bullying victimization group reported the poorest subjective well-being and the lowest resilience levels.

### 3.3. Poly-Bullying Victimization and Subjective Well-Being: Testing the Moderator Effect of Resilience

It was hypothesized that resilience would moderate the relation between poly-bullying victimization and subjective well-being. To examine this hypothesis, a hierarchical regression analysis was conducted with subjective well-being taken as the criterion variable. Although gender was not statistically significant in the previous analyses, it was inputted together with poly-bullying victimization and resilience in Model 1. In Model 2, the product of poly-bullying victimization and resilience was inputted to test the interaction effect. The results of Models 1,2 of the hierarchical regression analyses are reported in Table 3.

The results for Model 1 showed that gender had no significant effect on subjective well-being but revealed a significant main effect of poly-bullying victimization. That is, the participants who reported having experienced three or four forms of bullying were more likely to also report lower subjective well-being levels. Resilience was significantly and positively associated with subjective well-being. The participants who reported higher resilience levels also reported higher subjective well-being levels. The results in Model 2 showed that resilience moderated the association between poly-bullying victimization and subjective well-being, but the relation was very weak and accounted for only an additional 1% of variance in the participants’ subjective well-being (r^2^ change: 0.010, *p* < 0.01).

## 4. Discussion

The present study aimed to analyze the associations between poly-bullying victimization and reports of subjective well-being, and to explore whether resilience moderated the relation between these two variables.

Participants who had been victims of nearly all (three or four) forms of bullying victimization were classified as “poly-bullying victims”. A substantial proportion of the participants (16.9%) reported being victims of poly-bullying. The prevalence rates in this study were higher than those previously found, when the incidence of poly-victimization did not exceed 11% [9,30,62]. One explanation for these results could be the lack of consensus about how to measure poly-victimization, and how to establish who can be classified as a poly-victim [59]. Moreover, researchers such as Álvarez-Lister et al. [28] have pointed out that the number of victimization experiences that define poly-victims differs even in those studies using the same criteria. Prevalence rates for the different types of victimization were also higher in comparison to previous studies [12,13]. A possible reason for these higher rates may be that bullying self-reports could be filled by participants thinking on previous experiences during secondary school. Indeed, the survey was administered at the beginning of the academic year and bullying victimization rates were higher among students in the first year in comparison to students in second and third year.

Regarding the association between well-being and resilience among students exposed to several forms of bullying, the results fall in line with the expectations posed in Hypothesis 1. The results showed that the poly-bullying victimization group reported the poorest subjective well-being. The participants who reported having experienced three or four forms of bullying were more likely to also report lower subjective well-being levels. This finding coincides with previous research, which has shown that exposure to multiple forms of victimization has an impact on young people’s mental health, specifically their subjective well-being [33,45,49]. This suggests that poly-bullying victimization represents a set of potentially traumatic adverse experiences with stronger detrimental effects on well-being than experiencing only one form of victimization.

The regression analysis results indicated that resilience was significantly and positively associated with subjective well-being. The participants identified as non-victims reported better subjective well-being and higher resilience levels than victims. On the contrary, the poly-bullying victimization group reported the poorest subjective well-being and the lowest resilience level. These results confirmed Hypothesis 2 and fall in line with previous studies showing how young people who suffer victimization and poly-victimization had poorer mental health and lower resilience than those not victimized [58,59].

Finally, in line with previous studies [57,58,59,63], the results corroborated Hypothesis 3 and showed that resilience moderated the association between poly-bullying victimization and subjective well-being. However, this relation was very weak and represented only an additional 1% of variance in subjective well-being. Other researchers have also found a weak statistical power between resilience and bullying/cyberbullying [54,58]. In sum, our results indicated that resilience was related to subjective well-being and poly-bullying victimization, but not strong moderating effect of resilience on the relation was found between poly-bullying victimization and subjective well-being. This weak moderate effect could be explained by the fact that, although previous research has claimed that resilience is a potent protective factor, both in preventing experience with bullying and mitigating its effect [54], it is reasonable to consider that suffering bullying through multiple forms may have a detrimental effect over individuals’ levels of resilience. Resilience include stress moderators such as personal coping strategies (i.e., sense of personal agency and the repertoire of coping mechanisms they can utilize when faced with adversity), cognitive mechanisms (i.e., negative appraisals of the self and events in the world), and positive relationships (i.e., social support, peer acceptance and friendship networks) that can be affected by adversity [64]. In this sense, poly-bullying victimization could minimize youth’s personal and social resilience resources what in turn will affect their possibility to coping with the negative effects of bullying on their subjective well-being. Future research should assess resilience trajectories of youth exposed to multiple forms of bullying victimization in order to better understand the potential protective effect of resilience over negative mental health outcomes.

Nevertheless, conceptual discrepancies in the scientific community about the resilience concept make it difficult to evaluate and compare the research and operationalization results of the construct for measurement purposes [55]. The present results suggest that it would be important to consider more complex models of resilience (beyond considering resilience to be a personal trait) that contemplate both individual and contextual variables more broadly. It would also be necessary to explore a wider range of psychosocial variables, such as depression, self-esteem, self-control, internal locus of control, cohesion and family communication or positive school experiences, because they may mediate in relations among resilience, poly-bullying victimization and subjective well-being [58,65]. Additionally, future research should analyze resilience as a process and about the way that youths develop, interiorize and use their resilience resources in bullying victimization episodes, which differs between youths who recover well and whose subjective well-being is not affected by the victimization they suffered and those who suffer a detrimental effect on their well-being.

### Limitations and Directions for Future Research

When interpreting these findings, certain limitations in our study need to be considered. First, the study sample was confined to university students from a region of central Spain. Thus, it is difficult to establish whether the study findings are generalizable to Spanish population in other regions. It would be appropriate to replicate these findings in other settings. Second, given the survey’s cross-sectional nature, it did not allow us to establish causal relations between the analyzed variables. Longitudinal studies need to be conducted to begin examining possible causal relations. Third, this study included only an evaluation of positive mental health. It is important for future research to analyze other forms of internalization and externalization problems as consequences of poly-bullying, e.g., suicidal or aggressive behavior, anxiety or depression. It would also be necessary to replicate and validate these results by exploring a wider range of individuals and contextual psychosocial variables, e.g., self-esteem, self-efficacy, cohesion and family communication or positive school experiences. Fourth, it is also important to point out the inherent limitations of asking participants to self-report their experiences of victimization. Thus, collection data from multiple informants rather than relying exclusively on self-reports would be beneficial in future studies.

## 5. Conclusions

This study examined poly-bullying victimization in a sample of late adolescents. Those who were victims of poly-victimization reported more negative outcomes (worse subjective well-being and lower resilience levels) than the youths who experienced single forms of victimization or were not involved. Previous research and the present study reveal that negative outcomes from bullying victimization experiences differ according to the form of peer victimization and the number of experiences suffered. Although prevention and intervention efforts must deal with all victimization episodes, special attention should be paid to the youths suffering multiple forms of bullying victimization, and interventions need to be tailored to each form of victimization [33]. Therefore, prevention/intervention professionals should evaluate a wide range of victimization types when they are presented with a specific incident and should incorporate this knowledge into their practical responses. Although many national and international programs are being run to reduce bullying victimization in young people [66,67,68], it is still unclear what techniques and programs would be effective in reducing poly-bullying victimization due to the limited studies on this phenomenon. In order to reduce negative correlates on mental health, interventions to promote resilience by developing social and emotional competencies should be considered in addition to the aforementioned programs against bullying [69,70,71].

## Figures and Tables

**Table 1 ijerph-17-00590-t001:** Prevalence of each bullying victimization form and poly-bullying victims among Spanish university students.

Forms of Bullying	N = 1430
Non-victims	43.6% (624)
Physical bullying	17.7% (253)
Verbal bullying	29.5% (422)
Social bullying	46.1% (659)
Cyberbullying	17.7% (253)
Victims of a single form of bullying	24.1% (344)
Victims of two forms of bullying	15.4% (220)
Victims of three forms of bullying	11.6% (166)
Victims of four forms of bullying	5.3% (76)

**Table 2 ijerph-17-00590-t002:** Descriptive analyses of participants’ gender, well-being and resilience among the university students with or without different bullying victimization profiles.

Variables	Non-Victims	Victims of a Single Form of Bullying	Victims of Two Forms of Bullying	Poly-Victims (Three or Four Forms of Bullying)	η^2^
Female, %	63.0	67.7	65.9	59.5	
Subjective Well-being, M (SDs)	5.29 (0.89) ^a^	5.14 (0.81) ^b^	5.11 (0.89) ^c^	4.94 (0.81) ^d^	0.033
Resilience, M (SDs)	4.25 (0.88) ^a^	4.14 (0.84) ^b^	3.96 (0.97) ^c^	3.72 (1.03) ^d^	0.019

Note: the line mean values with different subscripts are significantly different. a > b > c > d.

**Table 3 ijerph-17-00590-t003:** Linear regression analyses examining the associations of resilience and bullying victimization and relatedness to subjective well-being as the criterion.

Variables	Model 1			Model 2		
	B	SE_B_	β	B	SE_B_	β
Gender ^a^	0.051	0.042	0.027	0.050	0.05	−0.07
Poly-bullying victimization	−0.080	0.016	−0.108 ***	−0.281	0.095	−0.380 ***
Resilience	0.562	0.080	0.542 ***	0.519	0.030	0.500 ***
Interaction: Resilience x poly-bullying victimization				0.040	0.018	0.273 **
R^2^ (Adj. R^2^)	0.339 (0.337)			0.349 (0.344)		
∆R^2^	0.339			0.010		
*F*	145.74 ***			69.07 ***		

Note. N = 1430. ^a^ 0 = female; 1 = male; ** *p* < 0.01; *** *p* < 0.001.

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
