# Peer review of "Relations among Poly-Bullying Victimization, Subjective Well-Being and Resilience in a Sample of Late Adolescents"

_ijerph, 2020, doi:10.3390/ijerph17020590_

Round 1

Reviewer 1 Report

The article supports previous research indicating that poly-bullying victimization results in poor subjective well-being and low resilience levels.  It also found a weak moderating effect of resilience on subjective well-being.  The literature review does a nice job of summarizing the literature and the methodology is sound.  One consideration that the authors may want to discuss in their results/discussion/conclusions is the possibility that resilience may be impacted by victimization.  The results indicate that poly-bullying victimization is significantly associated with lower resilience, yet there is no discussion on why this may be the case.  Without this discussion, it potentially reads as if people who have lower resilience are more likely to be poly-bullied. 

30-32: Could use a little elaboration on “changes in experienced social relationships, intense emotions, and biological changes”

71-73: Define Hickman’s violence measures (at least the lifetime of a broad category)

78: “more severe” instead of “severer”

94: Fix “That is to say, positive mental health is related to, but in a different form, mental illness” (currently says “from”)

96: “significantly” not “significant”

132-134:  “..as either cognitive-individual and social-interpersonal…” Either/or or both, needs clarified

148-149:  “found resilience” and “childhood depression”

153-157: What is the difference between “young people” and “late adolescents?” It reads that the intention of the paper is to replicate findings (awkward) among late adolescents, but needs to add a bit more to explain why this research is a contribution.

Author Response

The article supports previous research indicating that poly-bullying victimization results in poor subjective well-being and low resilience levels.  It also found a weak moderating effect of resilience on subjective well-being.  The literature review does a nice job of summarizing the literature and the methodology is sound.  One consideration that the authors may want to discuss in their results/discussion/conclusions is the possibility that resilience may be impacted by victimization.  The results indicate that poly-bullying victimization is significantly associated with lower resilience, yet there is no discussion on why this may be the case.  Without this discussion, it potentially reads as if people who have lower resilience are more likely to be poly-bullied. 

Authors’ answer: thank you very much for you comprehensive and interesting comments. We have now discuss the possibility that resilience may be impacted by victimization. We have added a new paragraph in lines 513-525.

30-32: Could use a little elaboration on “changes in experienced social relationships, intense emotions, and biological changes”

 Authors’ answer: thank you, we have elaborated that sentence.

71-73: Define Hickman’s violence measures (at least the lifetime of a broad category)

 Authors’ answer: thank you, we have defined “lifetime of a broad category”.

78: “more severe” instead of “severer”

 Authors’ answer: thank you, we have changed the word.

94: Fix “That is to say, positive mental health is related to, but in a different form, mental illness” (currently says “from”)

 Authors’ answer: thank you, we have changed the word.

 96: “significantly” not “significant”

  Authors’ answer: thank you, we have changed the word.

132-134:  “..as either cognitive-individual and social-interpersonal…” Either/or or both, needs clarified

  Authors’ answer: thank you, we have changed the word.

148-149:  “found resilience” and “childhood depression”

  Authors’ answer: thank you, we have changed the word.

153-157: What is the difference between “young people” and “late adolescents?” It reads that the intention of the paper is to replicate findings (awkward) among late adolescents, but needs to add a bit more to explain why this research is a contribution.

 Authors’ answer: thank you for your suggestion, we have now clarified the importance of the study in lines 270-274.

Reviewer 2 Report

Dear editor and authros,

Thank you for the opportunity to review this manuscript. I found the article  very interesting and concise and that it makes a contribution by covering a very importante topic. I particularly like that the authors have approached the topic from the poly victimization theory, and have also added two important variables SWB and Resilience, particularly the latter one as the authors state is a topic that has been understudied perhaps for the lack of consistency in its definition. All in all the authors propose a clear and concise study, with solid methods and building upon a great trajectory from their research team. Therefore I think the article would be ready for publication and that it will make a great contribution to the field.

I have only a few remarks/comments/suggestion that I would like the authors to address. Please do not be discouraged by the amount of points/comments (8) I am making, as all of them are minor remarks and suggestions, which I hope might help to improve a little this great manuscript.

Abstract

Point 1. I think it woudl be best to indicate the specific number of self-reports completed rather than "a number"…You can shorten that sentence that way. Perhaps you could indicate in the abstract a rate of prevalence of the problem and add a sort of conclusion. I think the word limit in this journal is quite high and the abstract is a important selling point of the manuscript for other scholars to read, so I could try to round it a bit more if possible and if you find it apprpiate.

Introduction

Point 2. The introduction properly defines, addresses and reviews the variables under study, I would add a study or two covering how often or usual (perhaps prevalence rates) is face-to-face bullying in university (see connection with Point 8). Maybe and misunderstood but I don’t know if the prevalence rates from line 44 to 51 refer to college students or adolescents, if they do refer to college students or late adolescents please state it clearly.

The aims and hypotheses are clearly laid out and refer back to the literature mentioned in the review.

Design

Point 3. If I am not mistaken, I think you mean to say that the study design is cross-sectional. I think the sentence is awkwardly phrased as Analyses themselves are any nature. (a regression can be used with cross-sectional or longitudinal data). Please correct me if I am wrong

Participants

Point 4. I leave wondering if this are first year undergrad students or, also from 2nd and 3rd year. I guess the authors took the decision to eliminate those who were older than 23 years old, or did this happened by chance? I also wonder why did the authros choose to not include some more information such as which studies they were enrolled on (Humanities, social sciences, psychology?) I found the description of the participants sufficient but particularly short.

Instruments

Point 5. I would ask the authors to be a bit more consistent when presenting the assessment tools, please provide the dimensions, time-framed asked ( very important in the case of bullying) number of items, response categories and anchors, and range of scores for all the instruments.

Results

The statistical analyses are appropriate and are adequately laid out. I have only two minor comments on this regard.

Point 6. I do not like to mix different statistics in the same table. In the case of table 2 I would delete the first row and present the chi square analyses in text. And leave table 2 just for the ANOVA. This is obviously just a minor suggestion and please feel free to leave it as it is.

Point 7. It is not necessary to duplicate the results from table 3 in the text.

Discussion

Point 8. Going back to Point 2…this seems such a high prevalence rate for bullying in university. Are these rates normal, I would like to see a little bit more on prevalence rates in University students on the introduction and then compared to yours. I find it disturbing that up to a 17.7% of our students have been victims of physical aggressions…could it be that they are referring to their institute experiences?

Very minor mistakes/typos 

Review sentence in lines 94-95. Perhaps different form? Or different from?

From line 233 to 238 the italics in the statistical symbols are missing as is the super index in the chi squared.    

My apologies for being so annoying about this but I think the new ICTs are not so new anymore, and that  “In recent years, a new form of bullying has emerged” is more appropriate for a 2008 definition than one from 2020.

Author Response

Dear editor and authors,

Thank you for the opportunity to review this manuscript. I found the article  very interesting and concise and that it makes a contribution by covering a very importante topic. I particularly like that the authors have approached the topic from the poly victimization theory, and have also added two important variables SWB and Resilience, particularly the latter one as the authors state is a topic that has been understudied perhaps for the lack of consistency in its definition. All in all the authors propose a clear and concise study, with solid methods and building upon a great trajectory from their research team. Therefore I think the article would be ready for publication and that it will make a great contribution to the field.

I have only a few remarks/comments/suggestion that I would like the authors to address. Please do not be discouraged by the amount of points/comments (8) I am making, as all of them are minor remarks and suggestions, which I hope might help to improve a little this great manuscript.

 Authors’ answer: thank you for your kind comments and your suggestions. They helped us to improve the description of the study.

Abstract

Point 1. I think it woudl be best to indicate the specific number of self-reports completed rather than "a number"…You can shorten that sentence that way. Perhaps you could indicate in the abstract a rate of prevalence of the problem and add a sort of conclusion. I think the word limit in this journal is quite high and the abstract is a important selling point of the manuscript for other scholars to read, so I could try to round it a bit more if possible and if you find it appropriate.

 Authors’ answer: following your suggestion we have added more information in the abstract.

Introduction

Point 2. The introduction properly defines, addresses and reviews the variables under study, I would add a study or two covering how often or usual (perhaps prevalence rates) is face-to-face bullying in university (see connection with Point 8). Maybe and misunderstood but I don’t know if the prevalence rates from line 44 to 51 refer to college students or adolescents, if they do refer to college students or late adolescents please state it clearly.

 Authors’ answer: thank you for your suggestion, we have now clarified the samples of the studies and we have added studies analyzing prevalence in college students to comment later in the discussion the differences found.

Design

Point 3. If I am not mistaken, I think you mean to say that the study design is cross-sectional. I think the sentence is awkwardly phrased as Analyses themselves are any nature. (a regression can be used with cross-sectional or longitudinal data). Please correct me if I am wrong

Authors’ answer: Thank you. Following your suggestion and comments from reviewer 3 we have re-elaborated the design subsection.

Participants

Point 4. I leave wondering if this are first year undergrad students or, also from 2nd and 3rd year. I guess the authors took the decision to eliminate those who were older than 23 years old, or did this happened by chance? I also wonder why did the authros choose to not include some more information such as which studies they were enrolled on (Humanities, social sciences, psychology?) I found the description of the participants sufficient but particularly short.

Authors’ answer: thank you. Following you suggestion and comments from reviewer 3 we have included more information about the procedure. We collected data from year 1 to 3.

Instruments

Point 5. I would ask the authors to be a bit more consistent when presenting the assessment tools, please provide the dimensions, time-framed asked ( very important in the case of bullying) number of items, response categories and anchors, and range of scores for all the instruments.

Authors’ answer: sorry about that. We have followed your suggestions and we have re-elaborated the information offered in the description of the instruments.

Results

The statistical analyses are appropriate and are adequately laid out. I have only two minor comments on this regard.

Point 6. I do not like to mix different statistics in the same table. In the case of table 2 I would delete the first row and present the chi square analyses in text. And leave table 2 just for the ANOVA. This is obviously just a minor suggestion and please feel free to leave it as it is.

 Authors’ answer: thank you for your suggestion. We have decided to leave the table as it is because trying to describe the results in the test may be more confusing to readers.

Point 7. It is not necessary to duplicate the results from table 3 in the text.

 Authors’ answer: thank you for your suggestion. We have eliminated data from the text.  

Discussion

Point 8. Going back to Point 2…this seems such a high prevalence rate for bullying in university. Are these rates normal, I would like to see a little bit more on prevalence rates in University students on the introduction and then compared to yours. I find it disturbing that up to a 17.7% of our students have been victims of physical aggressions…could it be that they are referring to their institute experiences?

  Authors’ answer: thank you for your suggestion, we have now included comparative studies. Prevalence results called our attention too. We think you are probably right and we have discussed it given that the higher prevalence rates were obtained in year 1.

Very minor mistakes/typos 

Review sentence in lines 94-95. Perhaps different form? Or different from?

From line 233 to 238 the italics in the statistical symbols are missing as is the super index in the chi squared.    

My apologies for being so annoying about this but I think the new ICTs are not so new anymore, and that  “In recent years, a new form of bullying has emerged” is more appropriate for a 2008 definition than one from 2020.

 Authors’ answer: thank you very much. It was helpful and you are right abouth the new ICTs. We have to erase that type of comment from our minds although sometimes seem to be write in stone.

Reviewer 3 Report

It is a interesting and well executed study. However, the manucript suffer some small weakness that should be addressed before it can be accepted for publication. 

Introduction:

Line 85-86 "The primary aim of the present study was to examine the relations among poly-bullying victimization (experiencing multiple forms of peer bullying), resilience and subjective well-being." From here, the authors should continue with some clarifications of poly-bullying victimization, what forms of bullying (verbal, physical) are included in their definition, which perpetrators are included, are syblings included as peers? and places (in school, out school, cyber)? This will make it clear for the readers in following literature review sections when other terms appear, e.g. Line 105, what is 'social bullying'? is it different from 'peer bullying'? Line 107, what is 'relational bullying', does it include 'peer bullying'?    

The manuscript needs a good round of English languages editing. Below are just some examples. 

The sentence of lines 56-57 sounds strange and please consider revision "Bullying victimization is also associated with mental health treatment and counseling [20]."

The sentence of lines 94-95 needs revision "That is to say, positive mental health is related to, but in a different from, mental illness [34]." 

The sentence of lines 125-126 I suggest add a word "Although research has shown that victimization experiences have short-term and long-term NEGATIVE effects on mental health, not all victims experience the same outcomes."

The sentence of lines 132-135, "The factors contributing to resilience have been conceptualized as either both cognitive-individual (self-regulation, planning, executive functioning, problem-solving skills and self-efficacy) and social-interpersonal (social skills, social support and quality of family relationships) [49]."

The sentence of lines 137-139, "In a meta-analysis, resilience was found to correlate negatively with negative affect, anxiety and depression, but positively with positive indicators of mental health, positive affect and 138 life satisfaction."

The H1, H2 and H3 between lines 164-171, in my opinion, can be deleted as they are un-neccessary after the clear statements of the a), b) and c) puroposes of the study. 

2. Method: 

The sentence between lines 174-176 "Cross-sectional analyses were performed with the data Data used in this study are from a non-probabilistic sample formed by containing 1430 undergraduate students from a Spanish university located in central Spain with 175 approximately 23000 students." The authors should continue this by providing information of sampling procedue, how the sample of 1430 were selected from a university with 23000 students.  

2.2 Measurement variables and instruments

Line 181, self-reported bullying victimization, please provide the exact wordings of questions asked in the survey, are the bullying experienced in the last 3 months retricted only between peers? and who are counted as peers? It will be good to have a table with descriptions of all measuring items. 

2.4 Analysis Plan

lines 214-218, I don't understand the reason of using hierarchical regression analysis as the data are cross-sectional and on one-level (individual level) only. 

3. Results

Table 1 title should more precise as the sample are not exactly 'Spanish youths', they are special as 'Spanish university students'

Table 2 title should also be more precise as it reports results of the whole sample of university students, not only 'among the late adolescent groups'.  

Table 3 title says 'Regression analyses' not hierarchical regression, as the results presented, it appears can be done with multiple, simple linear regression analysis function by SPSS, to introduce interaction in model 2. In any case, the authors need to justify the use of hierarchical regression on single level data. 

My reading stops after this section so I won't be able provide comments on the later sections. However, I recommend a thorough language editing after the authors revised the manuscript, before re-submission. 

Author Response

It is a interesting and well executed study. However, the manucript suffer some small weakness that should be addressed before it can be accepted for publication. 

Authors’ answer: thank you for your kind consideration and your useful review.

Introduction:

Line 85-86 "The primary aim of the present study was to examine the relations among poly-bullying victimization (experiencing multiple forms of peer bullying), resilience and subjective well-being." From here, the authors should continue with some clarifications of poly-bullying victimization, what forms of bullying (verbal, physical) are included in their definition, which perpetrators are included, are syblings included as peers? and places (in school, out school, cyber)? This will make it clear for the readers in following literature review sections when other terms appear, e.g. Line 105, what is 'social bullying'? is it different from 'peer bullying'? Line 107, what is 'relational bullying', does it include 'peer bullying'?    

The manuscript needs a good round of English languages editing. Below are just some examples. 

The sentence of lines 56-57 sounds strange and please consider revision "Bullying victimization is also associated with mental health treatment and counseling [20]."

The sentence of lines 94-95 needs revision "That is to say, positive mental health is related to, but in a different from, mental illness [34]." 

The sentence of lines 125-126 I suggest add a word "Although research has shown that victimization experiences have short-term and long-term NEGATIVE effects on mental health, not all victims experience the same outcomes."

The sentence of lines 132-135, "The factors contributing to resilience have been conceptualized as either both cognitive-individual (self-regulation, planning, executive functioning, problem-solving skills and self-efficacy) and social-interpersonal (social skills, social support and quality of family relationships) [49]."

The sentence of lines 137-139, "In a meta-analysis, resilience was found to correlate negatively with negative affect, anxiety and depression, but positively with positive indicators of mental health, positive affect and 138 life satisfaction."

Authors’s answer: thank you for you thorough review. The manuscript has been reviewed by a native English speaker, and we have followed all of your suggestion.

The H1, H2 and H3 between lines 164-171, in my opinion, can be deleted as they are un-neccessary after the clear statements of the a), b) and c) puroposes of the study. 

Authors’ answer: thank you for your suggestions. We have decided to leave the hypothesis as they are because we think are useful to understand comment in the discussion section.

Method: 

The sentence between lines 174-176 "Cross-sectional analyses were performed with the data Data used in this study are from a non-probabilistic sample formed by containing 1430 undergraduate students from a Spanish university located in central Spain with 175 approximately 23000 students." The authors should continue this by providing information of sampling procedue, how the sample of 1430 were selected from a university with 23000 students.  

Authors’ answer: following your suggestions and aso review 2 comments we have re-written the design and participants subsections.

2.2 Measurement variables and instruments

Line 181, self-reported bullying victimization, please provide the exact wordings of questions asked in the survey, are the bullying experienced in the last 3 months retricted only between peers? and who are counted as peers? It will be good to have a table with descriptions of all measuring items. 

Authors’ answer: thank you for you suggestion. Description of the instrument has been changed following your advice.

2.4 Analysis Plan

lines 214-218, I don't understand the reason of using hierarchical regression analysis as the data are cross-sectional and on one-level (individual level) only. 

Authors’ answer: thank you. That was a mistake written the article. We conducted a linear regression in the SPSS software. We have made appropriate changes in the description.

Results

Table 1 title should more precise as the sample are not exactly 'Spanish youths', they are special as 'Spanish university students'

Table 2 title should also be more precise as it reports results of the whole sample of university students, not only 'among the late adolescent groups'.  

Authors’ answer: thank you. We have rewritten the title of the tables.

Table 3 title says 'Regression analyses' not hierarchical regression, as the results presented, it appears can be done with multiple, simple linear regression analysis function by SPSS, to introduce interaction in model 2. In any case, the authors need to justify the use of hierarchical regression on single level data. 

Authors’ answer: thank you. That was a mistake written the article. We conducted a linear regression in the SPSS software. We have made appropriate changes in the description.